# Plant tissue analysis as a tool for predicting fertiliser needs for low cyanogenic glucoside levels in cassava roots: An assessment of its possible use

**Matema L. E. Imakumbili**[1]*, **Ernest Semu**[1], **Johnson M. R. Semoka**[1], **Adebayo Abass**[2], **Geoffrey Mkamilo**[3]

**1** Department of Soils and Geological Sciences, Sokoine University of Agriculture, Morogoro, Tanzania, **2** The International Institute of Tropical Agriculture, Dar es Salaam, Tanzania, **3** Roots and Tubers Department, Naliendele Agricultural Research Institute, Mtwara, Tanzania

* imakumbili@gmail.com

**Data Availability Statement:** All relevant data are within the manuscript and its Supporting Information files.

## Abstract

The use of plant tissue analysis as a tool for attaining low cyanogenic glucoside levels in cassava roots, has hardly been investigated. Just as the quality of crops is improved through the use of plant tissue analysis, the same can probably be done to consistently attain the lowest possible cyanogenic glucoside levels in cassava roots. High levels of cyanogenic glucosides in consumed fresh cassava roots or in their products have the potential of causing cyanide intoxication, hence the need to lower them. An experiment was thus conducted to assess the occurrence of meaningful relationships between plant nutritional status and cyanogenic glucoside production in cassava roots. Total hydrogen cyanide (HCN) levels in cassava roots were used to assess cyanogenic glucoside production. Using NPK fertiliser application to induce changes in plant nutritional status, the main objective of the study was investigated using the following sub-objectives; (1) to determine the effects of increased NPK fertiliser application on cassava root HCN levels; (2) and to show the occurrence of relationships between changes in nutrient levels in plant 'indicator tissue' and HCN levels in cassava roots. The study was a field experiment laid out as a split-plot in a randomized complete block design with three replicates. It was repeated in two consecutive years, with soil nutrient deficiencies only being corrected in the second year. The varieties *Salanga*, *Kalinda*, *Supa* and *Kiroba* were used in the experiment, while the NPK fertiliser treatments included; a control with no fertiliser applied; a moderate NPK treatment (50 kg N + 10 kg P + 50 kg K /ha); and a high NPK treatment (100 kg N + 25 kg P + 100 kg K /ha). A potassium only treatment (50 kg K/ha) was also included, but mainly for comparison. The root HCN levels of *Salanga*, *Kalinda* and *Kiroba* were significantly influenced by NPK fertiliser application in at least one of the two field experiments, while those of *Supa* remained uninfluenced. Changes in plant nutritional status in response to fertiliser application were thus shown to influence cyanogenic glucoside production. The results of the multiple linear regression analysis for the first field experiment, generally showed that the root HCN levels of some cassava varieties could have been 'reduced' by decreasing concentrations of nitrogen,

**Funding:** This work was made possible with funding from the Alliance for a Green Revolution in Africa (AGRA) (https://agra.org/) through its Soil Health Program (SHP). AGRA had provided MLEI a scholarship to study at Sokoine University of Agriculture (AGRA Grant number 2009 SHP 027). Partial funding had also been given to MLEI for cassava cyanide analysis by Dr J.H. Bradbury from the Australian National University (ANU) (http://www.anu.edu.au/). Funding was also separately provided by the Bill and Melinda Gates Foundation (https://www.gatesfoundation.org/) to cover the cost of publication; on the basis of their support to the AGRA SHP (Grant number OPP48790). The funders had no role in study design, data collection and analysis, decision to publish, or preparation of the manuscript.

**Competing interests:** The authors have declared that no competing interests exist.

potassium and magnesium in plants, or by improving plant calcium concentrations along with NPK fertiliser application. However, in the second field experiment (with corrected soil nutrient deficiencies) the regression analysis generally showed that the root HCN levels of some cassava varieties could have been 'reduced' by improving either one or a combination of the nutrients phosphorous, zinc and potassium in plants along with NPK fertiliser application. Although the results obtained in the two experiments had been contradicting due to slight differences in how they were conducted, the study had nonetheless demonstrated the occurrence of meaningful relationships between plant nutritional status and cyanogenic glucoside production; confirming the possible use of plant tissue analysis in predicting fertiliser needs for the consistent attainment of low cyanogenic glucosides in cassava roots.

## Introduction

Plant tissue analysis plays an important role in the improvement of crop yields and also in the enhancement of desirable crop quality characteristics. It has a diagnosis role, where it is used to confirm or dismiss the presence of limiting nutrients in soils that are highlighted by visual symptoms on plants; it also has a monitoring role, which ensures that growing crops always have adequate nutrients for consistent optimal growth; and it lastly has a supporting role, where it is used along with soil test results to make fertiliser recommendations [1]. The advantage of plant tissue analysis is that it reveals the actual nutritional status of plants, which may at times tell a different story from what is expected from pre-existing soil nutrient conditions. The invaluable information given by plant tissue analysis helps growers to quickly and more accurately identify whether plant nutritional problems emanate from a poor supply of nutrients or from the presence of other growth limiting factors.

Like with many other crops, most research on cassava (*Manihot esculenta* Crantz) involving plant tissue analysis, has mainly focused on its use for achieving better plant growth and root yields [2–5]. The use of plant tissue analysis as a tool for decreasing levels of cyanogenic glucosides in cassava, has hence hardly been investigated. Low cyanogenic glucoside levels are desirable in edible parts of cassava for the improvement of a crop quality characteristic related to its safe consumption. Although feared to be still toxic, total hydrogen cyanide (HCN) levels (a measure the cyanogenic glucoside content) of less than 50 mg/kg in fresh cassava roots are currently recommended as safe for consumption [6]. Research on cyanogenic glucosides in cassava has mainly focused on examining the influence of fertiliser application on cyanogenic glucoside content without linking the observations to plant nutritional status. The results obtained, particularly with the application of NPK (nitrogen, phosphorous and potassium) fertilisers have been varied; some studies have reported no effects on cassava root cyanogenic glucoside contents with NPK fertiliser application [7–9], while others have reported reductions [9] and one study even suggested the occurrence of increased root cyanogenic glucosides with NPK fertiliser application [10]. Plant tissue analysis could thus probably help to consistently produce cassava roots with low cyanogenic glucoside levels, if it can be used to develop fertiliser recommendations aimed at decreasing cyanogenic glucosides in cassava roots.

In its supportive role of formulating fertiliser recommendations, plant tissue analysis makes use of relationships between plant nutritional status and plant growth or yields. Relationships between plant nutrient concentrations in 'indicator tissue' and cyanogenic glucosides in cassava, hence need to be understood if cyanogenic glucoside production is to be effectively controlled with fertiliser application. Nutrient concentrations in plant 'indicator tissue' are used

because they are best correlated to plant growth and yields; they thus give the best representation of a plants nutritional status. Now, although fertiliser application in cassava production may be questioned, given the ability of cassava to remain productive under conditions of low soil fertility, the following arguments may help to justify its use or learning from its use. Firstly, whether fertilisers are applied or not applied, growing cassava plants are still influenced by soil nutrient supply which in turn affects plant nutritional status. It is changes in plant nutritional status that influence metabolic processes in cassava, like those involving the production of cyanogenic glucosides. Fertilisers only influence soil nutrient supply and they are used to carefully control soil nutrient supply. Fertilisers can thus be useful for learning how plant nutritional status affects cyanogenic glucoside production over a range of soil nutrient conditions. Secondly, while some may only support fertiliser application for cassava yield improvement, the applied fertilisers would still affect cyanogenic glucoside production through their effects on plant nutritional status.

The objective of this study, was to assess the occurrence of meaningful relationships between plant nutritional status and cyanogenic glucoside production in cassava roots. However, since the occurrence of relationships between plant nutritional status and cyanogenic glucoside production can be either highlighted indirectly by the effects of fertiliser application on cyanogenic glucoside production, or more directly through the evidence of relationships between changes in plant nutrient concentrations and cyanogenic glucoside production, the main objective of the study had to be fulfilled using two sub-objectives. Using increased NPK fertiliser application to induce changes in plant nutritional status, the first sub-objective was thus to determine the effects of increased NPK fertiliser application on the root HCN levels of cassava. While the second sub-objective was to show the occurrence of relationships between changes in nutrient concentrations in plant 'indicator tissue' and HCN levels in roots of cassava supplied with increased applications of NPK fertiliser. With this, the first hypothesis tested was that 'cassava *root HCN levels are not influenced by increased applications of NPK fertiliser*' and the second hypothesis tested was that '*root HCN levels in cassava supplied with increased levels of NPK fertiliser are not influenced by changes in nutrient concentrations in indicator tissue*.

## Materials and methods

### The field experiment

**Location.** A field experiment was conducted to investigate the effects of increased levels of NPK fertiliser application on root HCN levels and the relationships between nutrient concentrations in plant 'indicator tissue' and root HCN levels. The experiment had been repeated in two consecutive years. Different but closely located sites with no fertiliser application history at Naliendele Agricultural Research Institute (NARI) were used for the field experiment. The experimental sites were located on the Eastern Makonde Plateau in Mtwara rural district in Mtwara region of Tanzania. The district lies in the coastal agro-ecological zone 2 (C2) and is characterised by monomodal rainfall with a short growing season of 3 to 4½ months [11,12]. The rainy season begins in November/December and ends in April/May. The total rainfall in the district ranges from 800 to 1000 mm/year and has unreliable onset dates, including a characteristic 3 to 4 weeks long mid-season dry period (a seasonal interruption) normally experienced in February [11,13]. The district experiences mean minimum and maximum temperatures of 19 to 23 $^{o}$C and 29 to 31 $^{o}$C, respectively [13] and 79 to 87% relative humidity [11].

The first field experiment (Year 1) (S 10˚22'56", E 40˚10'00" at 133 m above sea level) was established in the last week of January 2014 and was ended in the first week of January 2015; it

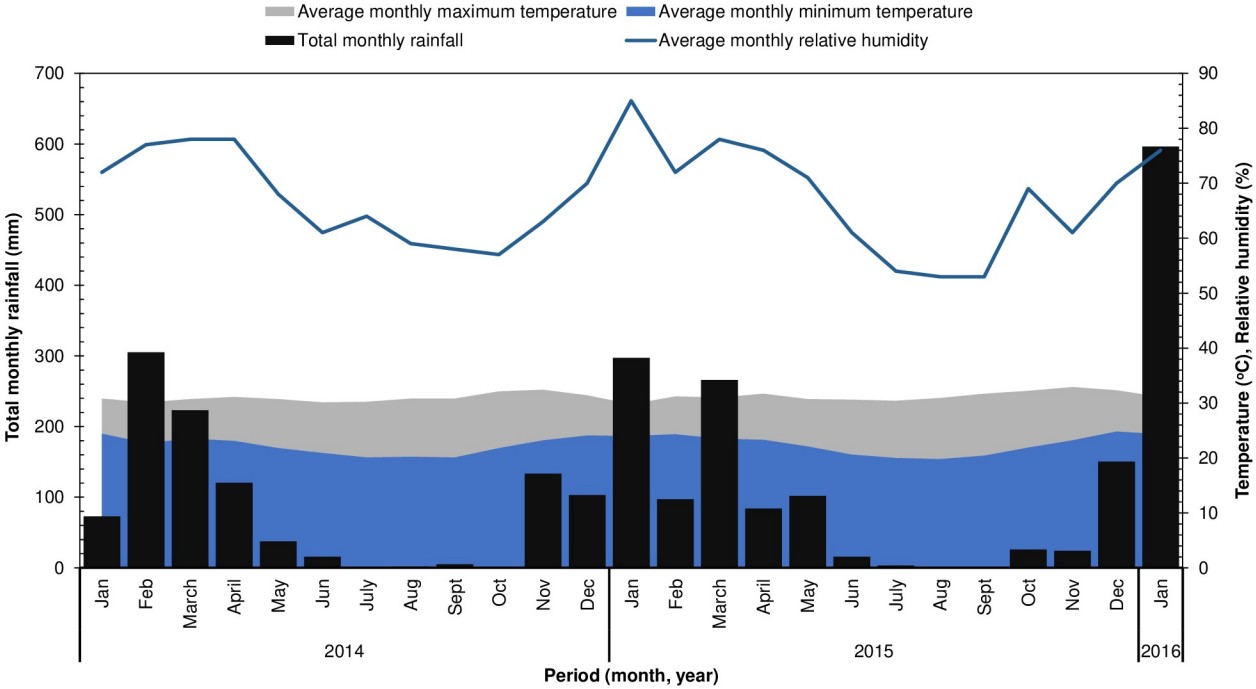

**Fig 1. Mean monthly rainfall, relative humidity, and maximum and minimum temperatures during the field experiments.**

was hence carried out for about 11 months. The second field experiment (Year 2) (S 10˚23'03", E 40˚09'49" at 158 m above sea level) was established in the last week of February 2015 and was ended in the first week of February 2016; it thus carried out for about 11 months, as well. The two field experiments were both conducted under rain-fed conditions. The mean monthly rainfall received and the mean monthly relative humidity and temperatures experienced by cassava plants during the two growing periods are shown in Fig 1.

Before planting, a 500 g composite soil sample had been collected from each site from a depth of 0 to 20 cm for soil analysis [https://dx.doi.org/10.17504/protocols.io.x72frqe]. The soils were analysed for organic carbon (OC), soil reaction (pH), total N, available P, available K, exchangeable calcium (Ca), exchangeable magnesium (Mg), available sulphur (S), zinc (Zn), copper (Cu) and iron (Fe) and for their soil texture. The soil analysis procedures were carried out as follows: pH was determined in $H_2O$ using a 1:1 soil to water ratio; OC using the Walkley and Black method; N by micro-Kjeldahl digestion; P by the Bray No. 1 method; Sulphate-S by the calcium phosphate extraction solution; available K, exchangeable Ca and Mg using 1 N ammonium acetate extracting solution; extractable Zn, Cu and Fe using diethylene-triaminepentaacetic acid (DTPA); and soil texture by the hydrometer method [14]. Table 1 shows the soil analysis results that were obtained.

The soils from the field experimental sites in Year 1 (85.87% sand, 11.26% clay, 2.87% silt) and Year 2 (86.37% sand, 11.26% clay, 2.37% silt) were both found to be loamy sands [19]. Soils in Mtwara region are generally classified as Ferralic Cambisols [11]; they are predominantly sandy and of low soil fertility [20,21].

**Experimental design.** The field experiment was a 4×4 factorial combination laid out as a split-plot in a randomized complete block design. Each block was replicated three times. Four cassava varieties and four fertiliser treatments were used in the experiment. Varieties were assigned to the main plots, while the fertiliser treatments were assigned to the sub-plots. Out of the four cassava varieties, one variety was an improved cassava variety called *Kiroba*, while the

**Table 1. Soil chemical characteristics for the field experimental sites in Years 1 and 2.**

| Parameter | Year 1 | | Year 2 | | Medium range/Critical level | Reference |
|---|---|---|---|---|---|---|
| | Value | Status | Value | Status | | |
| pH | 5.40 | m | 6.10 | m | 4.5–7.0 | [15] |
| OC (%) | 0.27 | vl | 0.53 | vl | 4.0–10.0 | [16] |
| N (%) | 0.04 | vl | 0.05 | l | 0.20–0.50 | [16] |
| P (mg/kg) | 1.49 | vl | 2.24 | l | < 4.2 | [17] |
| K (cmol/kg) | 0.12 | l | 0.03 | vl | 0.15–0.25 | [15] |
| Ca (cmol/kg) | 2.25 | m | 2.51 | m | 1.0–5.0 | [15] |
| Mg (cmol/kg) | 0.12 | vl | 0.41 | m | 0.40–1.00 | [15] |
| S (mg/kg) | 3.33 | l | 1.27 | l | < 6.0 | [16] |
| Zn (mg/kg) | 0.16 | vl | 0.31 | vl | 1.0–3.0 | [18] |
| Cu (mg/kg) | trace | vl | 0.13 | l | 0.3–0.8 | [18] |
| Fe (mg/kg) | 9.81 | h | 9.40 | h | 4.0–6.0 | [18] |

Where vl, l, m and h stand for very low, low, medium and high levels of each soil chemical characteristic.

other three were all local cassava varieties and were namely; *Supa*, *Salanga* and *Kalinda*. The local varieties were collected from Kitangari village (S 10˚39'01", E 39˚20'01") in Newala district, which is also located in Mtwara region, whereas *Kiroba* was collected from NARI. The stem cuttings for the local cassava varieties had been collected from farmers' crop fields. Written permission to conduct any research activities in relation to the study had initially been obtained from the regional government administrative offices in Mtwara region before the local cassava stem cuttings could be collected from farmers. Farmer consent was however only given verbally. Staff from NARI had assisted with all activities involving cassava stem cutting collection. All activities at farm level were conducted in the presence of government officers working from Newala district or with their knowledge.

The fertiliser treatments consisted of a control treatment ($N_0P_0K_0$) with no applied fertiliser; a sole K fertiliser treatment ($N_0P_0K_1$) with K applied at a rate of 50 kg K/ha; a moderate NPK fertiliser treatment ($N_1P_1K_1$) applied at a rate of 50 kg N + 10 kg P + 50 kg K /ha; and a high NPK fertiliser treatment ($N_2P_2K_2$) applied at a rate of 100 kg N + 25 kg P + 100 kg K /ha. The field experiments each had two other NPK fertiliser treatments but these were not included in the present study, as they did not either supply N, P and K at the same time or because their rates were not close equal increments of the selected NPK fertiliser treatments. The sole K fertiliser treatment was only included for comparison, given the well-known reducing effects of sole K fertiliser application on cyanogenic glucoside production in cassava; it was however excluded from some key analysis. Urea ($CO(NH_2)_2$), triple super phosphate (TSP) ($Ca(H_2PO_4)_2.H_2O$) and muriate of potash (KCl) were used to supply the N, P and K in the NPK fertiliser treatments. Urea and KCl were applied in two split applications, while the TSP was applied in one single application. The TSP was applied at planting together with one-third of the urea and KCl and the remaining two-thirds of the urea and KCl were applied at 2 months after planting (MAP). The fertiliser was banded near the cassava stem cuttings.

**Planting and crop management.** Each sub-plot was 6 m × 6 m in size and had a total of 36 plants, with 16 plants in the net (effective) plot. The cassava stem cuttings were planted at an inclined positon on the flat (no ridges used), using a spacing of 1 m × 1 m (plant population of 10 000 plants/ha). The field was maintained weed free for the first 3 MAP using hand-hoe weeding. This was achieved by weeding the field twice, first at just before 2 MAP and then at 3 MAP. A third and fourth weeding had however also been carried out at 6 MAP and between 9

and 10 MAP. Unlike in Year 2, the field experiment in Year 1 had been carried out without correcting soil nutrient deficiencies. This was actually because of the delay in getting the soil analysis results; it however brought out useful insights and was taken as something to learn from. The effects of NPK fertiliser application were hence examined using two scenarios; with and without the correction of soil nutrient deficiencies.

The soils at the experimental site in Year 2 were corrected for S, Zn and Cu. Magnesium sulphate was used to correct the S deficiency; it was applied at a rate of 20 kg S/ha [22] at 2 MAP together with the second application of urea and MOP. The deficiency in Zn was corrected using a 1% foliar solution of a product called YaraVita Zintrac (700 g Zn/L, as ZnO), while the Cu deficiency was corrected using a 0.05% foliar solution of YaraVita Coptrac (500 g Cu/L, as CuO). Two foliar applications of Zn and Cu were made; the first at 2 MAP and the second at 3 MAP. The insecticide Dursban was used to control insect pests; it was mixed with the foliar fertiliser solutions each time before spraying.

## The pot experiment

A pot experiment had also been conducted to get a preliminary analysis of the effects of NPK fertiliser application on cyanogenic glucoside production in cassava. The pot experiment was established on the 24th October 2015 and was ended on the 25th January 2016. It was hence carried out for a period of approximately for 3 months. The pot experiment was conducted at Sokoine University of Agriculture (SUA) (S 6°51'13", E 37°39'26") which is located in Morogoro district in Morogoro region of Tanzania. It was a 2×4 factorial combination laid out using the randomised complete block design (RCBD) using five replications. Only two out of the four varieties used in the field experiment had however been included in the pot experiment; namely the varieties *Supa* and *Salanga*. The same fertiliser treatments used in the field experiment were also used in the pot experiment. They had however been converted to pot based rates with the amount of nutrients applied expressed in milligrams per kilogram soil. The fertiliser treatments included; a control treatment ($N_0P_0K_0$) with no fertiliser applied; a sole K fertiliser treatment ($N_0P_0K_1$) with K fertiliser applied at a rate of 25 mg K/kg; a moderate NPK fertiliser treatment ($N_1P_1K_1$) applied at a rate of 25 mg N + 5 mg P + 25 mg K /kg; and a high NPK fertiliser treatment ($N_2P_2K_2$) applied at a rate of 50 mg N + 13 mg P + 50 mg K /kg.

Previously rooted cassava plantlets (shoots) had been used for the pot experiment and not 20 cm long cassava stem cuttings [https://dx.doi.org/10.17504/protocols.io.z9cf92w]. The use of rooted cassava plantlets enables the observation of immediate effects from applied fertilisers; this is because there is no interference from large nutrient reserves in plantlets. Each pot had been filled with an air-dry mass of soil equivalent to 5 kg of oven-dry soil. The soil used as potting medium was a loamy sand (86.05% sand, 11.46% clay and 2.49% silt) [19] collected from Soga village (S 6°49'54", E 38°51'49") in Kibaha district. The district is located in the Coast region of Tanzania. Soils in Kibaha district are also predominantly Ferralic Cambisols with an inherently low soil fertility [11,23]. The soil had been collected from 10 selected locations on a farmer's crop field from a 0 to 20 cm depth [https://dx.doi.org/10.17504/protocols.io.2eygbfw] [24,25]. The soil had been analysed to reveal nutrient deficiencies; the results of the soil analysis are given in Table 2.

The TSP and KCl had all been added and thoroughly mixed into the soil of each pot before planting [https://dx.doi.org/10.17504/protocols.io.4engtde]. The urea was however applied in solution form to already established cassava plants [https://dx.doi.org/10.17504/protocols.io.4ifgubn] [25]; it was applied in two split applications, with one-third and two-thirds of it being respectively applied at 2 and 6 weeks after planting (WAP). Deficiencies of Mg and S (Table 2) had been corrected by applying magnesium sulphate ($MgSO_4.7H_2O$) to each pot at a rate of 25

**Table 2. Chemical characteristics of soil used in the pot experiment.**

| Parameter | Pot experiment | | Medium range/Critical level | Reference |
|---|---|---|---|---|
| | Value | Status | | |
| pH | 5.80 | m | 4.5–7.0 | [15] |
| OC (%) | 0.35 | vl | 4.0–10.0 | [16] |
| N (%) | 0.06 | vl | 0.20–0.50 | [16] |
| P (mg/kg) | 3.54 | l | < 4.2 | [17] |
| K (cmol/kg) | 0.14 | l | 0.15–0.25 | [15] |
| Ca (cmol/kg) | 3.04 | m | 1.0–5.0 | [15] |
| Mg (cmol/kg) | 0.08 | vl | 0.40–1.00 | [15] |
| S (mg/kg) | 1.27 | l | < 6.0 | [16] |
| Zn (mg/kg) | 0.82 | l | 1.0–3.0 | [18] |
| Cu (mg/kg) | 0.70 | m | 0.3–0.8 | [18] |
| Fe (mg/kg) | 25.12 | vh | 4.0–6.0 | [18] |

Where vl, l, m and h stand for very low, low, medium and high levels of each soil chemical characteristic.

mg Mg/kg; simultaneously adding S at a rate of 32.5 mg/kg. The $MgSO_4.7H_2O$ was all applied before planting together with the TSP and KCl. The deficiency of Zn was corrected using a 1% solution of YaraVita Zintrac; it was sprayed on cassava plants at 1 and 2 MAP. Insect pests had also been controlled using the insecticide Dursban. The mean minimum and maximum temperatures in the screen house were respectively 23 and 33°C. All plants had been kept well-watered throughout the experiment.

Note that even though the pot experiment was conducted for 3 months, it would be advisable to conduct cassava pot experiments for at least 2 months or for more than 3 months. This is because root development would have already began in plants at about 3 MAP. In the present experiment, root development was observed to be uneven with both unfertilised and fertilised plants having roots or no roots produced on them at 3 MAP; this may introduce experimental variability on some sensitive growth and plant quality characteristics.

Permission had been sought from the district offices in Kibaha district before collecting the potting soil from the farmer's field. The permission given was not in written form; a letter from us requesting for help from the district was however left at the district office and we were only made to sign the visitor's book to indicate the purpose of our visit. The district office had arranged for the soil collection and all activities had been done in their presence and with their involvement. The farmer had only given verbal consent for his field to be sampled.

## Data collected

**Cyanogenic glucoside contents of cassava leaves and roots.** In order to assess the effects of NPK fertiliser application on cyanogenic glucoside production, changes in the HCN levels of cassava leaves and roots (tuber parenchyma) were determined from the pot and field experiments, respectively. In the field experiments, four plants were selected from the net plot of each sub-plot and three roots were then collected from each plant for the determination of HCN levels in cassava roots for each treatment [https://dx.doi.org/10.17504/protocols.io.ydxfs7n] [26]. In the pot experiment leaf sampling had been carried out from five plants. The first fully-expanded leaf from the top of a cassava plant plus two leaves below it were picked from each plant during sample collection [https://dx.doi.org/10.17504/protocols.io.2dbga2n] [26]. Leaf sampling was carried out at 92 days after planting in the early morning hours while ambient temperatures were still low. The picrate paper method was used to determine the

HCN levels of cassava roots [https://dx.doi.org/10.17504/protocols.io.2emgbc6] and leaves [https://dx.doi.org/10.17504/protocols.io.2dxga7n] [27,28]. Spectrophotometer readings for both leaf and root HCN levels had been determined and used in the analysis. Colour chart readings had however also been determined and these readings were used for *Supa* in Year 1, as its spectrophotometer readings had been affected by the use of a faulty blank. All results were reported on a fresh weight (fw) basis.

**Plant nutrient concentrations and nutritional status.** Plant nutrient concentrations had only been determined in the two field experiments and not in the pot experiment. Relationships between nutrient concentrations in 'indicator tissue' and root HCN levels had thus been based on field data. Nutrient concentrations in the youngest fully expanded leaves (YFEL) (without petioles) of cassava plants between 3 to 4 MAP are the best indicator tissue for the assessment of plant nutritional status for cassava [15]. The YFEL's were thus collected from all cassava plants in the net plot of each sub-plot at 3 to 4 MAP and placed in khaki paper bags. The collected leaves still in their paper bags were immediately placed in an air-forced oven and left to dry to a constant weight at 70 $^{\circ}$C. Once dry the leaf samples were finely ground and mixed in a porcelain mortar with a pestle; care was taken to avoid contamination between samples. The ground leaves were placed in plastic bags, which were then tightly sealed. Each plastic bag with its leaf sample was then placed in properly labelled khaki paper bags to keep them away from light. The leaf samples were stored in a cool dark place in the laboratory until the time for plant tissue analysis.

Dry ashing had been used to determine nutrient concentrations in the sampled leaves. The ash obtained for each treatment had been dissolved in 6 N hydrochloric acid (HCl) before determining concentrations of P, K, Ca, Mg, and Zn in the leaf samples [18]. Total N concentrations in the YFEL's was determined separately using the Kjeldahl method [16]. The nutrient concentrations that were determined were then rated according to their adequacy for cassava production to give the nutritional status of the cassava plants under each treatment [22].

## Statistical analysis

The effects of NPK fertiliser application on leaf and root HCN levels in the pot and field experiments, were respectively determined using a two-way and three-way Analysis of Variance (ANOVA) and mean separation was carried out using the Tukey's test at the 5% probability level [29]. The HCN responses to fertiliser application had also been mathematically modelled using regression analysis, in order to explain the observed responses [30]. Multiple linear regression analysis using the stepwise approach was additionally carried out to show the occurrence of relationships between root HCN levels and changes in the nutrient concentrations of the YFEL's in response to NPK fertiliser application [31]. The data had been cleaned to remove outliers and all statistical analyses were carried out using GenStat Edition 14.

## Results and discussion

### Effects of NPK fertiliser application

The Year by Variety by Fertiliser (Y×V×F) interaction effect from the three-way ANOVA carried out to determine how root HCN levels in the cassava varieties had been influenced by NPK fertiliser application in the two field experiments was significant (p = 0.018). This showed differences in how root HCN levels had been influenced in the cassava varieties in Years 1 and 2. The data was thus split by year and a two-way ANOVA was carried out to investigate how root HCN levels in the cassava varieties had been separately influenced in each year. The tables obtained for the two-way ANOVA's are shown in Table 3, together with the table for the two-way ANOVA for the pot experiment.

**Table 3. Two-way ANOVA tables for the effects of NPK fertiliser application on leaf HCN levels from the pot experiment and on root HCN levels from the field experiments in Years 1 and 2.**

| Source of variation | df | SS | MS | F | p-value | CV (%) |
|---|---|---|---|---|---|---|
| *Pot experiment* | | | | | | |
| Block | 4 | 5445.7 | 1361.4 | 3.18 | | 20.7 |
| Variety (V) | 1 | 50347.7 | 50347.7 | 117.54 | < 0.001 *** | |
| Fertiliser (F) | 3 | 55077.2 | 18359.1 | 42.86 | < 0.001 *** | |
| V×F | 3 | 26566.5 | 8855.5 | 20.67 | < 0.001 *** | |
| Residual | 25 | 10709.0 | 428.4 | | | |
| Total | 36 | 140034.0 | | | | |
| *Field experiment*: Year 1 | | | | | | |
| Block | 2 | 115.7 | 57.9 | 1.59 | | 12.1 |
| Variety (V) | 3 | 106440.1 | 35480.0 | 973.21 | < 0.001 *** | |
| Residual | 6 | 218.7 | 36.5 | 0.81 | | |
| Fertiliser (F) | 3 | 7597.7 | 2532.6 | 56.09 | < 0.001 *** | |
| V×F | 9 | 9412.7 | 1045.9 | 23.16 | < 0.001 *** | |
| Residual | 22 | 993.3 | 45.2 | | | |
| Total | 45 | 106343.3 | | | | |
| *Field experiment*: Year 2 | | | | | | |
| Block | 2 | 49.0 | 24.5 | 0.06 | | 15.4 |
| Variety (V) | 3 | 108903.2 | 36301.1 | 83.91 | < 0.001 *** | |
| Residual | 6 | 2595.7 | 432.6 | 4.19 | | |
| Fertiliser (F) | 3 | 728.5 | 242.8 | 2.35 | 0.100 NS | |
| V×F | 9 | 4139.7 | 460.0 | 4.46 | 0.002 ** | |
| Residual | 22 | 2271.4 | 103.2 | | | |
| Total | 45 | 116032.2 | | | | |

Where; df, SS, MS, F, p-value and CV stand for degree of freedom, sum of squares, mean square, computed F, probability value and coefficient of variation, respectively.

*** Significant at p < 0.001

** significant at p < 0.01

* significant at p < 0.05 and NS is not significant (p > 0.05).

The significant V×F interaction effect (p < 0.001) obtained in each experiment (Table 3) tells us that at least one cassava variety had responded differently to the effects of NPK fertiliser application in each experiment. How root HCN levels had been influenced by NPK fertiliser application in each cassava variety in all three experiments, was hence separately determined using a one-way ANOVA. The results obtained from all experiments are shown in Table 4, where it can be seen that fertiliser application had influenced all leaf HCN contents and only some root HCN contents in the cassava varieties. The occurrence of significant effects is an indication of the influence of changes in plant nutritional status (induced by NPK fertiliser application) on cyanogenic glucoside production in the cassava varieties. The effects of fertiliser application indirectly confirms the occurrence of relationships between plant nutritional status and cyanogenic glucoside production in cassava. The observed effects will be discussed in greater detail in the paragraphs that follow.

Table 4 shows that the leaf HCN levels of both *Salanga* and *Supa* had been significantly influenced by NPK fertiliser application in the pot experiment. However, unlike *Salanga*, no significant responses to NPK fertiliser application were correspondingly seen in the root HCN levels of *Supa* in any of the two field experiments. This shows that the effects of NPK fertiliser application on leaf HCN contents in cassava varieties, may not always translate to similar

**Table 4. HCN levels in leaves and roots of each variety in the pot and field experiments under the effects of increased NPK fertiliser application.**

| Variety | Fertiliser treatment | Pot experiment | | Field experiment | | | |
|---|---|---|---|---|---|---|---|
| | | | | Year 1 | | Year 2 | |
| | | Leaf HCN, fw | SE | Root HCN, fw | SE | Root HCN, fw | SE |
| | | (mg/kg) | (mg/kg) | (mg/kg) | (mg/kg) | (mg/kg) | (mg/kg) |
| Salanga | $N_0P_0K_0$ | 100.0$^{bc}$ | 10.0 | 108.7$^c$ | 7.3 | 142.4$^a$ | 10.4 |
| | $N_0P_0K_1$ | 66.9$^c$ | 7.0 | 99.0$^c$ | 4.5 | 131.7$^a$ | 12.4 |
| | $N_1P_1K_1$ | 141.9$^b$ | 13.7 | 147.3$^b$ | 7.7 | 132.5$^a$ | 22.5 |
| | $N_2P_2K_2$ | 233.6$^a$ | 18.2 | 188.8$^a$ | 2.0 | 175.6$^a$ | 11.8 |
| Kailnda† | $N_0P_0K_0$ | - | - | 30.2$^a$ | 0.0 | 32.0$^b$ | 3.6 |
| | $N_0P_0K_1$ | - | - | 22.3$^a$ | 2.4 | 47.4$^a$ | 3.2 |
| | $N_1P_1K_1$ | - | - | 44.0$^a$ | 6.4 | 53.2$^a$ | 7.6 |
| | $N_2P_2K_2$ | - | - | 37.3$^a$ | 2.4 | 39.8$^{ab}$ | 0.8 |
| Supa | $N_0P_0K_0$ | 47.8$^c$ | 9.2 | 17.3$^a$ | 2.7 | 17.7$^a$ | 2.2 |
| | $N_0P_0K_1$ | 54.4$^{bc}$ | 3.7 | 12.0$^a$ | 1.0 | 20.7$^a$ | 1.9 |
| | $N_1P_1K_1$ | 74.6$^{ab}$ | 3.6 | 16.7$^a$ | 0.7 | 18.4$^a$ | 2.2 |
| | $N_2P_2K_2$ | 84.9$^a$ | 7.9 | 16.3$^a$ | 1.5 | 22.8$^a$ | 2.1 |
| Kiroba† | $N_0P_0K_0$ | - | - | 32.7$^b$ | 1.9 | 54.4$^a$ | 4.1 |
| | $N_0P_0K_1$ | - | - | 32.3$^b$ | 3.8 | 63.1$^a$ | 6.4 |
| | $N_1P_1K_1$ | - | - | 31.7$^b$ | 0.3 | 55.0$^a$ | 6.5 |
| | $N_2P_2K_2$ | - | - | 54.0$^a$ | 2.0 | 51.2$^a$ | 1.9 |

†Kalinda and Kiroba were not included in the pot experiment. For each variety, means in the same column followed by the same lowercase letter are not significantly different at $p < 0.05$ using the Tukey's test. Leaf and root HCN levels were determined on a fresh weight (fw) basis. SE is the standard error of the mean. Where; $N_0P_0K_0$ = no fertiliser, $N_0P_0K_1$ = 50 kg K/ha, $N_1P_1K_1$ = 50 kg N + 10 kg P + 50 kg K /ha and $N_2P_2K_2$ = 100 kg N + 25 kg P + 100 kg K /ha for the field experiments or their equivalent rates in mg/kg for the pot experiment.

responses in root HCN contents. It should however be noted that the leaf and root HCN levels in the pot and field experiments had been determined on different plants and plant parts, which could have contributed to the different results observed for *Supa*. Cyanogenic glucoside contents of cassava however additionally reduce with plant age and may only stabilise after 10 to 12 MAP [32,33]. This makes plant age at harvest another possible reason for the different responses seen between the HCN contents of leaves and roots of *Supa* (3 and 11 MAP, respectively). On the other hand, similar to the observations made with *Salanga* in Year 1, another study also reported significant effects on the HCN contents of leaves and roots of field grown cassava at 9 MAP in response to NPK fertiliser application; the reported leaf and root HCN contents were on average 0.54 and 0.51 mg/100g fw (5.4 and 5.1 mg/kg fw), respectively [10].

Focusing on the two field experiments, the root HCN contents of *Salanga* and *Kiroba*, had only been significantly influenced by NPK fertiliser application in Year 1 and not in Year 2, whereas the root HCN contents of *Kalinda* had only been significantly influenced by NPK fertiliser application in Year 2 and not in Year 1 (Table 4). As already mentioned, other studies have similarly reported significant changes in cassava root HCN levels with NPK fertiliser application [10,34], while some have reported no responses [7,8,35]. It is possible that the non-response to NPK fertilisers by once responsive varieties had been due to the heavy rains received right before harvest (Fig 1); cyanogenic glucoside levels in cassava can be influenced by changes in environmental conditions right before harvest. The correction of soil nutrient deficiencies in Year 2 and their non-correction in Year 1 is however another possible reason. It is important to note that *Kalinda* was the only variety that had its root HCN levels influenced by NPK fertiliser application in Year 2 and that *Supa* was the only variety with its root HCN

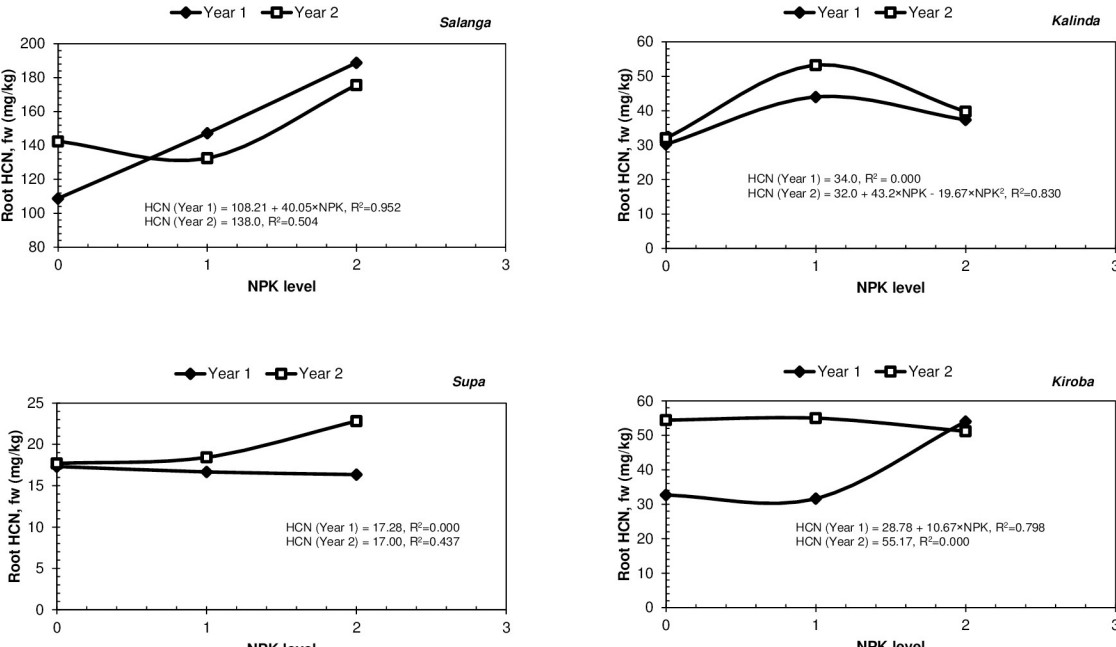

**Fig 2. Models of HCN responses to increased NPK fertiliser application for each cassava variety in the field experiments in Years 1 and 2.** Note that an ordinal scale was used to represent the fertiliser treatments and that; 0 = ($N_0P_0K_0$), 1 = $N_1P_1K_1$ and 2 = $N_2P_2K_2$. Where; $N_0P_0K_0$ = no fertiliser; $N_1P_1K_1$ = 50 kg N + 10 kg P + 50 kg K /ha; $N_2P_2K_2$ = 100 kg N + 25 kg P + 100 kg K /ha. Note that the sole K treatment ($N_0P_0K_1$) has not been included in these analyses.

content not influenced by NPK fertiliser application in both years. Fertiliser application may thus influence plant nutritional status in cassava varieties differently even under similar environmental conditions; resulting in varied effects on cyanogenic glucoside production.

From Table 4 it can additionally be seen that increased applications of NPK fertiliser, had resulted in heightened leaf HCN levels in both *Salanga* and *Supa* in the pot experiment. Increased NPK fertiliser application had likewise resulted in heightened HCN levels in roots of *Salanga* and *Kiroba* in the field experiment in Year 1. NPK fertiliser application had also heightened the root HCN levels of *Kalinda* in Year 2 but as revealed in the quadratic regression equation in Fig 2, the HCN levels of *Kalinda* had reached their highest limit with moderate applications of NPK fertiliser, with increased applications of NPK fertiliser only reducing its root HCN levels. It is however important to note that the ANOVA results in Table 4 showed no significant differences between the root HCN levels attained with moderate and high applications of NPK fertiliser in *Kalinda*. In contrast to *Kalinda*, the regression equations obtained for the effects of NPK fertiliser application on the root HCN levels of *Kiroba* and *Salanga* in Year 1 were both linear, showing that root HCN levels in the two varieties had not reached their highest (or lowest) limit with the applied NPK fertilisers (Fig 2). The regression equations obtained for the effects of NPK fertiliser application on the leaf HCN levels of *Salanga* and *Supa*, were also linear (Fig 3) and had thus not reached their highest (or lowest) limit with the applied NPK fertilisers, as well.

In contrast to the findings of the present study, other studies have often reported reductions in the HCN content of cassava with NPK fertiliser application (N, P and K supplied simultaneously) [36]. In one such study, root HCN levels of cassava were reported to reduce from 107.05 mg/kg fw when unfertilised to 77.49 mg/kg fw with high applications of NPK fertiliser at a rate of 100 kg N + 100 kg $P_2O_5$ + 100 kg $K_2O$ /ha (100 kg N + 44 kg P + 83 kg K /ha) [34].

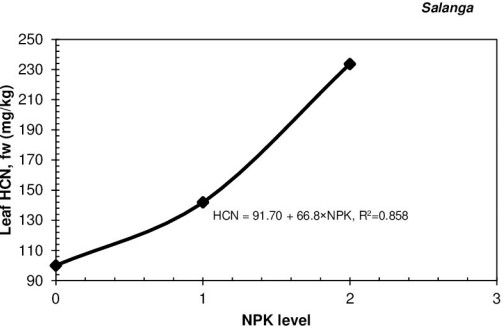

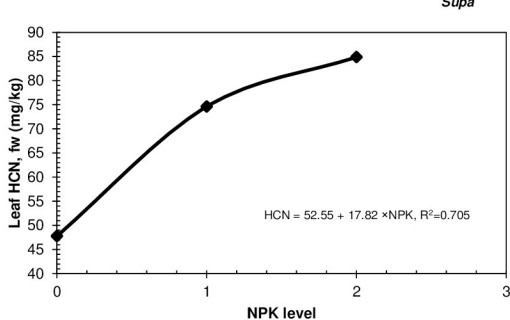

**Fig 3. Models of HCN responses to increased NPK fertiliser application for each cassava variety in the pot experiment.** Note that an ordinal scale was used to represent the fertiliser treatments and that; $0 = (N_0P_0K_0)$, $1 = N_1P_1K_1$ and $2 = N_2P_2K_2$. Where; $N_0P_0K_0 =$ no fertiliser; $N_1P_1K_1 = 25$ mg N + 5 mg P + 25 mg K /kg; $N_2P_2K_2 = 50$ mg N + 13 mg P + 50 mg K /kg. Note that the sole K treatment $(N_0P_0K_1)$ has not been included in these analyses.

In another such study, root HCN levels in cassava were reported to be 64.2 mg/kg on a dry weight (dw) basis under unfertilised conditions but had significantly reduced to 56.0 mg/kg dw with the application of NPK fertiliser at a rate of 60 kg N + 26 kg P + 50 kg K /ha [37]. Moderate applications of NPK fertiliser at a rate of 50 kg N + 21 kg P + 41 kg K /ha, were also reported to reduce HCN levels in cassava roots from 189 to 148 mg/kg fw, in yet another study [38].

Despite NPK fertilisers having an increasing effects on the HCN contents of cassava varieties in the present study, moderate applications of NPK fertiliser (50 kg N + 10 kg P + 50 kg K /ha) had mainly brought about lower HCN levels in leaves and roots in contrast to high applications of NPK fertiliser (100 kg N + 25 kg P + 100 kg K /ha) (Table 4). In *Kiroba*, moderate levels of NPK fertiliser were just as effective at reducing root HCN levels as sole K application in Year 1. As previously mentioned, sole K fertiliser application is often reported to have reducing effects on the HCN content of cassava [39,40]. However, just as observed in the present study when no effects were seen with NPK fertiliser application (Table 4), some studies have also reported no effects on root HCN contents with sole K fertiliser application [41]. Leaving *Salanga* and *Kiroba* unfertilised also proved to be at times just as effective as sole K fertiliser application at reducing leaf and/or root HCN levels in these cassava varieties. In Year 2, it was however seen that leaving *Kalinda* unfertilised was just as advantageous at reducing root HCN levels as high applications of NPK fertiliser. High applications of NPK fertiliser could thus be still useful for reducing the root HCN contents of these cassava varieties.

## Correlations between HCN levels in leaves and roots and between HCN levels of roots in Years 1 and 2

The leaf HCN levels of *Salanga* were significantly and positively correlated to the root HCN levels of this variety in both Year 1 (r = 0.795, p = 0.003) and Year 2 (r = 0.639, p = 0.034). The positive correlation coefficients show that the HCN contents of the leaves and roots of *Salanga* had both increased with increased applications of NPK fertiliser, although not significantly for root HCN contents in Year 2 (Table 4). On the contrary, the leaf HCN levels of *Supa* had shown no association with the root HCN levels it had obtained in Year 1 (r = 0.033, p = 0.920) and in Year 2 (r = 0.513, p = 0.088). Cyanogenic glucoside production was hence differently influenced in the leaves and roots of *Supa*. In agreement with these findings, one study also confirmed that not all varieties have significant correlations between their leaf and root HCN levels [42]. In another study, a significant positive correlation (r = 0.66, p = 0.0034) was also

reported between HCN contents in roots and leaves across four cultivars; the trend was however not consistent across all the sites where the experiment had been conducted [43]. Differences in environmental conditions hence also influence whether significant correlations are seen between the HCN contents of leaves and roots of cassava varieties. Non-significant relationships between HCN levels of leaves and roots were however expected, due to the different ages of cassava plants at leaf and root harvest.

Correlations between root HCN levels of cassava varieties in Year 1 with those obtained in Year 2 were however all not significant (Table 5), confirming that the response to NPK fertiliser application by each cassava variety had been completely different in the two field experiments. Differences in soil environmental conditions due to the correction and non-correction of soil nutrient deficiencies was a probable contributing factor for the absence of relationships between root HCN contents in Years 1 and 2.

## Relationships between changes in plant nutrient concentrations and root HCN levels

The results of the multiple linear regression analysis carried out to draw out relationships between plant nutrient concentrations in the YFEL's and the root HCN levels of each cassava variety in the field experiments in Years 1 and 2 are shown in Table 6. It is important to mention that the sole K treatment was excluded from the multiple linear regression analysis. The results hence show relationships with increased levels of NPK fertiliser application; from non-applied to the addition of moderate and finally high rates of NPK fertiliser. The significant relationships revealed by the regression analysis show that cyanogenic glucoside production was dependent on changes in plant nutritional status (Table 6); confirming the occurrence of relationships between plant nutritional status and cyanogenic glucoside production. The relationships are evidence of the strong dependency that plants have on their nutrient concentrations for the biosynthesis of quality determining organic compounds [44], like cyanogenic glucosides.

From Table 6 it can generally be seen that the root HCN contents of each cassava variety in the two field experiments, were likely to be influenced by different changes in their plant nutrient concentrations. As previously mentioned, the uncorrected nutrient deficiencies in Year 1 and the high rainfall received right before harvest in Year 2 might have brought about the observed differences in how root HCN contents were influenced in the two field experiments. In Table 6 it can as well be seen that the likely effects of changes in plant nutritional status on root HCN contents was not going to be restricted to varieties that had their root HCN contents significantly influenced by NPK fertiliser application (Table 4). It can further be seen that some varieties that had their root HCN contents influenced by NPK fertiliser application were not going to be further influenced by any changes in plant nutritional status. The results will now be discussed in greater detail below. Note that constant reference will also be made to the ANOVA results in Table 4.

**Table 5. Correlations between the HCN levels in roots of each variety in Years 1 and 2.**

| Variety | r | p-value |
|---|---|---|
| Salanga | 0.596 | 0.069 NS |
| Kalinda | 0.103 | 0.777 NS |
| Supa | -0.229 | 0.473 NS |
| Kiroba | -0.407 | 0.190 NS |

*** Significant at $p < 0.001$, ** significant at $p < 0.01$, * significant at $p < 0.05$ and NS is not significant ($p > 0.05$) using the Pearson correlation (two-tailed). Where r is the correlation coefficient.

**Table 6. Results of the multiple linear regression analysis showing relationships between nutrient concentrations in the YFEL's and the root HCN levels of cassava varieties in the two field experiments.**

| Field experiment | Variety | Nutrient | B | t | p-value | $R^2$ |
|---|---|---|---|---|---|---|
| Year 1 | *Salanga* | Ca | -4975.00 | -2.67 | 0.056 NS | 0.742 |
| | *Kalinda*[†] | - | - | - | - | - |
| | *Supa* | Ca | -115.20 | -3.02 | 0.023 * | 0.733 |
| | *Kiroba* | N | 13.23 | 7.53 | < 0.001 *** | 0.967 |
| | | K | 26.66 | 5.12 | 0.004 ** | |
| | | Mg | 613.00 | 3.75 | 0.013 * | |
| Year 2 | *Salanga*[†] | - | - | - | - | - |
| | *Kalinda* | P | -105.50 | -2.13 | 0.087 NS | 0.812 |
| | | Zn | -0.955 | -2.88 | 0.035 * | |
| | *Supa*[†] | - | - | - | - | - |
| | *Kiroba* | K | -42.4 | -2.97 | 0.021 * | 0.703 |

[†]Where a dash (-) indicates that no nutrient had a significant relationship with the root HCN content for that cassava variety.

*** Significant at $p < 0.001$

** significant at $p < 0.01$

* significant at $p < 0.05$ and NS is not significant ($p > 0.05$). Note that the sole K treatment was excluded from the multiple linear regression analysis.

The results of the multiple linear regression analysis shows that with the applied NPK fertilisers, a 1% increase in plant Ca concentrations in Year 1 would have helped reduce the root HCN levels of *Salanga* and *Supa* by 4975.00 and 115.20 times, respectively (Table 6). The very deficient plant Ca concentrations in *Salanga* and *Supa* (Table 7) support the increased demand for Ca. Calcium was in fact the most severely deficient plant nutrient in all the cassava varieties in Year 1, despite the soil at the experimental site in Year 1 having had adequate levels of Ca (Table 1). Root HCN levels of *Kalinda* and *Kiroba* were however unaffected by low plant Ca concentrations, probably because of their better adaptation to low concentrations of Ca. Cassava has been reported to have an outstanding ability to grow under growth limiting conditions of severe deficiencies of Ca in soils [45] and hence probably with severe deficiencies of Ca in plants. The uptake of Ca in all the varieties in Year 1, was probably limited by nutrient imbalances introduced by the non-correction of Mg, S, Zn and Cu in soils as the supply of N, P and K was increased. Calcium (and Mg) uptake is reported to be depressed by large additions of K fertilisers (like KCl) and ammonium-N containing fertilisers (like urea) to soils [46,47].

Regardless of not having been influenced by NPK fertiliser application in Year 1 (Table 4), the root HCN levels of *Supa* could thus have been reduced (and probably significantly) by increasing the concentrations of Ca in its plants. The root HCN contents of *Salanga* that had been increased by NPK fertiliser application, would also have been reduced if only the concentration of Ca had been increased in plants along with additions of NPK fertiliser. Improving Ca absorption would have helped alleviate nutrient stress due to low Ca concentrations in plants; consequently reducing the root HCN contents of the two cassava varieties. Increased applications of Ca through additions of up to 2000 kg/ha of lime (CaO) on acidic soils, were reported to be beneficial for decreasing root HCN levels in cassava from 54.8 to 40.8 mg/kg fw [48]. The HCN reducing effects of Ca could have resulted from improved concentrations of Ca in cassava plants.

Once again, but this time with *Kiroba* in Year 1, the multiple linear regression analysis had revealed that while keeping plant K and Mg concentrations unchanged, a 1% increase of N in plants, would have increased root HCN levels of *Kiroba* by 13.23 times (Table 6). Whereas, while keeping plant N and Mg concentrations the same, a 1% increase of K in plants, would

**Table 7. Nutrient concentrations in the youngest fully expanded leaves of cassava varieties under each NPK treatment in Year 1.**

| Variety | Fertiliser treatment | N | | P | | K | | Ca | | Mg | | Zn | |
|---|---|---|---|---|---|---|---|---|---|---|---|---|---|
| | | (%) | Status[‡] | (%)[*] | Status | (%) | Status | (%) | Status | (%) | Status | (ppm) | Status |
| Salanga | $N_0P_0K_0$ | 4.37 | d | 0.39 | s | 2.15 | h | 0.05 | vd | 0.21 | d | 40.63 | s |
| | $N_0P_0K_1$ | 3.99 | vd | 0.35 | d | 2.04 | h | 0.05 | vd | 0.21 | d | 40.13 | s |
| | $N_1P_1K_1$ | 5.58 | s | 0.42 | s | 2.11 | h | 0.04 | vd | 0.20 | d | 31.88 | d |
| | $N_2P_2K_2$ | 5.67 | s | 0.52 | h | 2.25 | h | 0.04 | vd | 0.21 | d | 34.05 | l |
| | CV (%) | 13.3 | | 13.9 | | 17.0 | | 6.8 | | 5.2 | | 12.5 | |
| Kalinda | $N_0P_0K_0$ | 3.96 | vd | 0.49 | s | 2.15 | h | 0.04 | vd | 0.20 | d | 44.88 | s |
| | $N_0P_0K_1$ | 3.84 | vd | 0.45 | s | 2.10 | h | 0.04 | vd | 0.22 | d | 46.13 | s |
| | $N_1P_1K_1$ | 5.61 | s | 0.43 | s | 2.33 | h | 0.03 | vd | 0.21 | d | 41.38 | s |
| | $N_2P_2K_2$ | 6.03 | h | 0.46 | s | 2.36 | h | 0.03 | vd | 0.20 | d | 39.38 | s |
| | CV (%) | 5.1 | | 20.9 | | 9.3 | | 16.2 | | 2.4 | | 6.1 | |
| Supa | $N_0P_0K_0$ | 3.56 | vd | 0.38 | l | 2.06 | h | 0.09 | vd | 0.22 | d | 42.38 | s |
| | $N_0P_0K_1$ | 3.63 | vd | 0.37 | l | 2.16 | h | 0.08 | vd | 0.21 | d | 48.92 | s |
| | $N_1P_1K_1$ | 4.50 | d | 0.32 | d | 2.28 | h | 0.07 | vd | 0.21 | d | 35.88 | s |
| | $N_2P_2K_2$ | 4.74 | d | 0.38 | l | 2.15 | h | 0.06 | vd | 0.20 | d | 37.38 | s |
| | CV (%) | 3.5 | | 11.4 | | 7.9 | | 13.9 | | 8.9 | | 18.5 | |
| Kiroba | $N_0P_0K_0$ | 4.04 | vd | 0.40 | s | 2.18 | h | 0.04 | vd | 0.18 | d | 39.88 | s |
| | $N_0P_0K_1$ | 3.75 | vd | 0.40 | s | 2.09 | h | 0.05 | vd | 0.18 | d | 36.38 | s |
| | $N_1P_1K_1$ | 4.51 | d | 0.36 | d | 2.12 | h | 0.04 | vd | 0.18 | d | 26.38 | d |
| | $N_2P_2K_2$ | 5.20 | s | 0.38 | s | 2.31 | h | 0.04 | vd | 0.19 | d | 32.13 | l |
| | CV (%) | 3.8 | | 15.0 | | 2.5 | | 15.9 | | 4.4 | | 9.8 | |

[‡]Where vd, d, l, s, h and t stand for very deficient, deficient, low, sufficient, high and toxic plant nutrient concentrations.

[*]Some values may appear similar due to rounding-off, but are different, hence their different plant nutritional status. *CV* is the coefficient of variation. Where; $N_0P_0K_0$ = no fertiliser, $N_0P_0K_1$ = 50 kg K/ha, $N_1P_1K_1$ = 50 kg N + 10 kg P + 50 kg K /ha and $N_2P_2K_2$ = 100 kg N + 25 kg P + 100 kg K /ha.

have increased the root HCN levels of *Kiroba* by 26.66 times. Still with *Kiroba* in Year 1, however this time while keeping plant concentrations of N and K unchanged, a 1% percent increase of Mg in plants, would have increased the root HCN levels of *Kiroba* by 613.00 times. The results indicate that increased concentrations of N, K and Mg in plants, would have had negative effects on cyanogenic glucoside production in roots of *Kiroba* in Year 1. The root HCN levels of *Kiroba* that were increased by NPK fertiliser application (Table 4), could hence have been much higher in Year 1 if concentrations of N, K and Mg had been further increased in plants. Higher concentrations of N in plants were going to be the single most main contributor of increased root HCN levels in *Kiroba*, followed by higher concentrations of K and then by higher concentrations of Mg in plants (Table 6).

It is easy to see why higher concentrations of N in plants could have increased the root HCN levels of *Kiroba* in Year 1 and this is because the concentration of N in plants had already reached high levels with the applied N fertilisers (Table 7). Higher concentrations of N in *Kiroba* plants, would hence have increased plant N concentrations to toxic levels which would have resulted in plant stress and consequently into higher root HCN levels. *Kiroba* was actually the only variety in which plant N concentrations had reached high levels with the applied NPK fertilisers in Year 1. An improved supply of N in soils is often reported to have increasing effects on cyanogenic glucoside production in cassava, particularly when it is supplied in high amounts and as the main nutrient in fertilisers [49,50]. Toxic levels of K could also have been easily reached in *Kiroba* in Year 1, if the concentration of K had been further increased in plants. *Kiroba* had high concentrations of K, but so did the other cassava varieties (Table 7).

Cyanogenic glucoside production in *Kiroba*, was hence probably most sensitive to high concentrations of plant K. As previously mentioned, the increased supply of K in soils and probably increased concentrations of K in plants, is often reported to reduce root HCN levels in cassava. Contrary to this, the present study shows that increased concentrations of K in cassava plants through additions of K fertilisers, could also have increasing effects on root HCN contents. Similar to the results of the present study, increased levels of K (0.03 to 1.9 cmol/kg) in soils, and probably in plants, were also observed to increase with root bitterness (and thus root HCN levels) in cassava [51]. Increased bitterness of cassava roots is often positively correlated to root HCN levels [52].

Moving to Mg, with its concentrations in plants having been less than adequate in *Kiroba* in Year 1, it was expected that increasing concentrations of Mg in plants would be beneficial for reducing root HCN levels. It is therefore not clear why improved plant Mg concentrations would have increased root HCN levels in *Kiroba*. A possible explanation is that increased plant concentrations of Mg in *Kiroba*, would have induced antagonistic nutrient interactions, like those preventing the absorption of other needed nutrients, particularly those that were inadequate in plants, like Ca. The close relationship that plant Mg concentrations have with the supply of Ca (and K) in soils [46], makes this a possible contributing factor. This reason, again points out to the effects of introduced nutrient imbalances in the field experiment in Year 1. However, another possible reason is that although rated as deficient, plant Mg concentrations could probably have been adequate for *Kiroba*, if it has an ability of utilising minimal levels of Mg. This can however be only confirmed by using plant Mg requirements specifically developed for *Kiroba* in this growing environment. An improved supply of Mg in soils has been reported to have reducing effects on cassava root HCN levels [34], similar to the effects of the increased supply of K in soils. On the hand, similar to the findings of the present study, increased soil pH (4.40 to 6.96), which was significantly positively associated with increased levels of Mg (r = 0.548, p-value < 0.001; ranging from 0.02 to 0.76 cmol/kg) and K (r = 0.343, p-value < 0.001; ranging from 0.02 to 0.32 cmol/kg) in soils, was reported to likely increase the occurrence of high root HCN levels in cassava produced in Mtwara region [21]. The concentrations of Mg in *Kiroba* could hence have been simply adequate or even high given their levels in soils (Table 1).

The significant results of the multiple linear regression analysis in Year 2, cannot be adequately discussed without first describing the changes that occurred to the nutrient concentrations of the YFEL's as a result of NPK fertiliser application. The nutrient concentrations obtained in the YFEL's in Year 2 are shown in Table 8, where it can be seen that N and Zn were the only nutrients that were mainly adequate in all cassava varieties. Nitrogen was adequate even in unfertilised plants, implying that the adequate levels of N in plants had been mainly been brought about by the correction of soil nutrient deficiencies. The occurrence of several nutrient deficiencies in plants (as shown in Table 8), despite their adequate supply, however points out to the occurrence of a 'dilution effect' [53]. A 'dilution effect' occurs when plants undergo rapid dry matter production that exceeds the rate of nutrient accumulation [53]. When factors that lead to a 'dilution effect' occur at the time of tissue sampling for plant analysis, it results in nutrient concentrations appearing to be lower than their actual levels in plants.

A 'dilution effect' occurs when a growth limiting factor is removed; similar to what was observed in the present study when soil nutrient deficiencies were corrected (Table 8). Most nutrients were thus adequate in plants in Year 2 and this was confirmed by their better growth, in contrast to the growth of plants in Year 1 (Table 9). It can thus be concluded that the differences between the results obtained in the two field experiments had been mainly due to the uncorrected nutrient differences in Year 1. This does not however rule out any effects that

**Table 8. Nutrient concentrations in the youngest fully expanded leaves of cassava varieties under each NPK treatment in Year 2.**

| Variety | Fertiliser treatment | N | | P | | K | | Ca | | Mg | | Zn | |
|---------|---------------------|-----|-----------|-----|--------|-----|--------|-----|--------|-----|--------|-------|--------|
| | | (%) | Status[‡] | (%) | Status | (%) | Status | (%) | Status | (%) | Status | (ppm) | Status |
| Salanga | $N_0P_0K_0$ | 5.35 | s | 0.28 | d | 1.00 | d | 0.30 | d | 0.17 | d | 47.74 | s |
| | $N_0P_0K_1$ | 5.47 | s | 0.29 | d | 1.00 | d | 0.31 | d | 0.17 | d | 46.64 | s |
| | $N_1P_1K_1$ | 5.55 | s | 0.29 | d | 0.97 | d | 0.26 | d | 0.17 | d | 42.24 | s |
| | $N_2P_2K_2$ | 5.88 | h | 0.34 | d | 1.07 | d | 0.28 | d | 0.16 | d | 35.64 | s |
| | CV (%) | 8.8 | | 14.7 | | 13.1 | | 12.5 | | 8.6 | | 16.0 | |
| Kalinda | $N_0P_0K_0$ | 5.14 | s | 0.29 | d | 1.15 | d | 0.31 | d | 0.19 | d | 52.14 | s |
| | $N_0P_0K_1$ | 5.07 | l | 0.27 | d | 1.08 | d | 0.33 | d | 0.18 | d | 49.94 | s |
| | $N_1P_1K_1$ | 5.15 | s | 0.27 | d | 1.17 | d | 0.32 | d | 0.16 | d | 41.14 | s |
| | $N_2P_2K_2$ | 5.52 | s | 0.32 | d | 1.23 | d | 0.33 | d | 0.17 | d | 44.44 | s |
| | CV (%) | 5.2 | | 8.3 | | 9.8 | | 5.8 | | 14.6 | | 14.9 | |
| Supa | $N_0P_0K_0$ | 5.04 | l | 0.28 | d | 1.06 | d | 0.29 | d | 0.16 | d | 38.94 | s |
| | $N_0P_0K_1$ | 5.05 | l | 0.26 | d | 1.04 | d | 0.30 | d | 0.16 | d | 40.04 | s |
| | $N_1P_1K_1$ | 5.37 | s | 0.32 | d | 1.16 | d | 0.26 | d | 0.16 | d | 35.64 | s |
| | $N_2P_2K_2$ | 5.51 | s | 0.32 | d | 1.15 | d | 0.28 | d | 0.15 | d | 33.44 | l |
| | CV (%) | 3.8 | | 8.4 | | 10.2 | | 12.8 | | 4.4 | | 15.9 | |
| Kiroba | $N_0P_0K_0$ | 5.30 | s | 0.27 | d | 1.08 | d | 0.32 | d | 0.14 | vd | 43.34 | s |
| | $N_0P_0K_1$ | 5.25 | s | 0.29 | d | 1.14 | s | 0.31 | d | 0.15 | d | 40.04 | s |
| | $N_1P_1K_1$ | 5.43 | s | 0.28 | d | 1.13 | d | 0.29 | d | 0.15 | vd | 27.94 | d |
| | $N_2P_2K_2$ | 5.61 | s | 0.26 | d | 1.02 | d | 0.33 | d | 0.15 | d | 35.64 | s |
| | CV (%) | 2.4 | | 16.0 | | 7.9 | | 7.1 | | 5.4 | | 16.3 | |

[‡]Where vd, d, l, s, h and t stand for very deficient, deficient, low, sufficient, high and toxic plant nutrient concentrations. CV is the coefficient of variation. Where; $N_0P_0K_0$ = no fertiliser, $N_0P_0K_1$ = 50 kg K/ha, $N_1P_1K_1$ = 50 kg N + 10 kg P + 50 kg K /ha and $N_2P_2K_2$ = 100 kg N + 25 kg P + 100 kg K /ha.

could have been introduced by the heavy rains experienced right before harvest in Year 2. The results of the multiple linear regression analysis were however not affected by the 'dilution effect', as they had still retained their trends. They are hence still reliable and shall now be discussed in detail in the paragraphs that follow.

The results of the multiple linear regression analysis in Table 6, shows that there were no significant relationships between the root HCN contents and plant nutrient concentrations of *Salanga* and *Supa* in Year 2; implying that no further changes in the nutritional status of plants of these varieties were needed to change their root HCN contents with the applied NPK fertilisers. The plant nutritional status of these varieties was hence in equilibrium with the rate at

**Table 9. Mean root dry matter contents, plant heights and stem diameters for each cassava variety at 11 MAP in Years 1 and 2.**

| Variety | Year 1 | | | Year 2 | | |
|---------|--------------|---------------|--------------------|--------------|---------------|-----------------|
| | Plant height | Stem diameter | Root DM[‡] content | Plant height | Stem diameter | Root DM content |
| | (cm) | (cm) | (%) | (cm) | (cm) | (cm) |
| Salanga | 212.5 | 2.1 | 27.4 | 292.0 | 2.4 | 22.6 |
| Kalinda | 168.6 | 1.9 | 27.4 | 280.0 | 2.3 | 29.4 |
| Supa | 156.2 | 1.8 | 28.5 | 292.4 | 2.4 | 33.1 |
| Kiroba | 132.0 | 1.6 | 28.3 | 225.8 | 2.2 | 32.0 |
| CV (%) | 15.6 | 11.9 | 5.7 | 11.7 | 9.7 | 9.6 |

[‡]Where DM stands for dry matter.

which they were producing cyanogenic glucosides. No significant relationships between root HCN contents and plant nutrient concentrations were also seen in *Kalinda* in Year 1. Back to Year 2, only *Kalinda* and *Kiroba* had significant relationships between their plant nutrient concentrations and root HCN levels. The regression analysis showed that while keeping plant Zn concentrations unchanged, a 1% increase in plant P concentrations would have reduced the root HCN levels of *Kalinda* by 105.50 times. In addition, with plant P concentrations being kept the same, a 1% increase in Zn would have reduced the root HCN levels of *Kalinda* by 0.955 times. The root HCN levels of *Kalinda* shown in Table 4 could hence have been much lower, if only the concentrations of P and Zn had been further increased in plants along with the applied NPK fertilisers. Increasing plant concentrations of Zn in *Kalinda* was slightly more critical for getting the root HCN levels of this variety reduced than increasing concentrations of P.

An improved supply of Zn in soils was reported to have reducing effects on the root HCN contents of cassava [48], on the other hand an improved supply of P in soils was reported to have no influence on the root HCN contents of cassava [50]. Increments of P and Zn had probably simultaneously occurred in plants with the improved supply of P and Zn in soils, in both studies. The present study reveals that P can also influence cyanogenic glucoside production in cassava, just like Zn. The better nutrition in Year 2 appeared to increase the demand for P and Zn in *Kalinda*, beyond their supplied amounts. Thus, in order to increase concentrations of P and Zn in *Kalinda* to their required levels; P should have been applied at a rate greater than 25 kg/ha and Zn should have instead been applied to soils at a rate of 10 to 20 kg/ha and not by foliar application [22]. Cassava has been reported to have a high demand for P [45], which could explain the increased demand for P by *Kalinda*. It is however unclear, why higher concentrations of Zn in plants could have helped reduce root HCN levels in *Kalinda* when Zn appeared to be mainly sufficient (Table 8). The deficiency of Zn in *Kalinda* could have however been masked by the Zn foliar applications, which could have only provided temporary relief from the deficiency. A possible explanation for the higher demand for Zn by *Kalinda*, could be its slightly higher variety specific demand for Zn (and P) but this can only be confirmed by using established plant nutritional requirements for *Kalinda* in this growing environment.

For *Kiroba*, increasing K in plants by 1% would have reduced root HCN levels by 42.4 times in Year 2 (Table 6). *Kiroba* appeared to have a higher demand for K in Year 2, which was again probably due to the better plant nutrition in the second field experiment. Although no significant effects had been seen with NPK fertiliser application in Year 2 for *Kiroba* (Table 4), the results of the multiple linear regression analysis showed that higher concentrations of K in plants along with NPK fertiliser application, could have helped lower (probably significantly) the root HCN levels obtained for this variety. Higher rates of K than what was applied (more than 100 kg K/ha) were hence probably needed to attain lower root HCN contents in *Kiroba*. Increasing the nutritional status of K in plants, would hence have had opposite effects on the root HCN content of *Kiroba* in the two field experiments. In Year 1, less plant K was needed to lower root HCN levels in *Kiroba*, while more K was needed to achieve this same effect in Year 2.

## Conclusions

Despite the differences between the results obtained in the two field experiments, due to the correction and non-correction of soil nutrient deficiencies, the study had still managed to demonstrate the occurrence of meaningful relationships between plant nutritional status and cyanogenic glucoside production in cassava roots. This was firstly shown through the observed

effects of NPK fertiliser application on the root HCN contents of various cassava varieties, which indirectly confirmed a response to changes in plant nutritional status, hence giving evidence of relationships between plant nutritional status and cyanogenic glucoside production; and it was secondly directly shown through the relationships observed between changes in nutrient concentrations of 'indicator tissue' and root HCN levels of various cassava varieties. The occurrence of meaningful relationships between plant nutritional status and cyanogenic glucoside production hence demonstrated that plant tissue analysis can be a useful tool for predicting fertiliser needs for the production of cassava roots with low cyanogenic glucoside contents.

The relationships qualify as being meaningful because of their close similarity with relationships that exist between plant nutritional status and plant growth or yields. For instance, very high and deficient plant nutrient concentrations both gave negative (increasing) effects on cyanogenic glucoside production, much like the negative (reducing) effects they have on plant growth and yields. In addition, much like what is observed with plant growth and yields, cyanogenic glucoside production also varies amongst various cassava varieties and under different growing environments. Different graphical response models (curves) representing relationships between plant nutrient concentrations and root HCN contents, are hence expected for different cassava varieties and in different growing environments, much like what is also seen with yields and plant growth.

While the use of fertilisers may not be economical for most farmers, the use of plant tissue analysis can still play a role in aiding the identification of cassava varieties that can be grown by farmers for safe consumption right at harvest. The root HCN contents of such varieties should only be minimally affected by changes in plant nutritional status; enabling them to consistently maintain low HCN contents. Furthermore, to fully exploit the benefits of plant tissue analysis, plant nutrient requirements that achieve both high yields and reduced cyanogenic glucoside levels in cassava need to be developed, given the importance of both characteristics for food and nutrition security.

## Supporting information

**S1 Dataset. Cassava root HCN levels, plant nutrient concentrations and growth characteristics.**
(XLSX)

**S2 Dataset. Leaf HCN levels.**
(XLSX)

**S1 Text. Soil profile description.**
(DOCX)

## Acknowledgments

As authors we thank all the funders for making this study possible; without them the study would not have occurred. We are also grateful for the insights received on cassava cyanide analysis from Dr J.H. Bradbury from the Australian National University and for the useful insights that we had also got on how to conduct cassava experiments from Dr R.H. Howeler; we surely had help from the best experts. We are additionally thankful to the staff at the Roots and Tubers Department at Naliendele Agriculture Research Institute and the staff at the Horticulture and Soil Science Departments at Sokoine University of Agriculture; their technical support definitely eased the research work. Another thank you goes to all the government officers that helped with the study in one way or another. Last but not least we thank the farmers that

provided the cassava stem cuttings and the potting soil used in the study; research would surely be impossible without the invaluable input we get from farmers.

## Author Contributions

**Conceptualization:** Matema L. E. Imakumbili.

**Data curation:** Matema L. E. Imakumbili.

**Formal analysis:** Matema L. E. Imakumbili.

**Funding acquisition:** Matema L. E. Imakumbili.

**Investigation:** Matema L. E. Imakumbili.

**Methodology:** Matema L. E. Imakumbili, Ernest Semu, Johnson M. R. Semoka, Geoffrey Mkamilo.

**Supervision:** Ernest Semu, Johnson M. R. Semoka, Adebayo Abass, Geoffrey Mkamilo.

**Validation:** Matema L. E. Imakumbili, Ernest Semu, Johnson M. R. Semoka.

**Visualization:** Matema L. E. Imakumbili.

**Writing – original draft:** Matema L. E. Imakumbili.

**Writing – review & editing:** Matema L. E. Imakumbili, Ernest Semu, Johnson M. R. Semoka.

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
