## [Decision Letter · Decision Letter 0]

6 Nov 2019

PONE-D-19-21677

Plant tissue analysis as tool for reducing cyanogenic glucoside contents in roots of cassava supplied with balanced NPK fertilisers

PLOS ONE

Dear %TITLE% Imakumbili,

Thank you for submitting your manuscript to PLOS ONE. After careful consideration, we feel that it has merit but does not fully meet PLOS ONE’s publication criteria as it currently stands. Therefore, we invite you to submit a revised version of the manuscript that addresses the points raised during the review process.

We would appreciate receiving your revised manuscript by Dec 21 2019 11:59PM. To enhance the reproducibility of your results, we recommend that if applicable you deposit your laboratory protocols in protocols.io, where a protocol can be assigned its own identifier (DOI) such that it can be cited independently in the future. For instructions see: http://journals.plos.org/plosone/s/submission-guidelines#loc-laboratory-protocols

We look forward to receiving your revised manuscript.

Kind regards,

Paulo H. Pagliari

Academic Editor

PLOS ONE

Journal Requirements:

Reviewers' comments:

Reviewer's Responses to Questions

**Comments to the Author**

1. Is the manuscript technically sound, and do the data support the conclusions?

Reviewer #1: Yes

Reviewer #2: Partly

2. Has the statistical analysis been performed appropriately and rigorously? 

Reviewer #1: Yes

Reviewer #2: Yes

3. Have the authors made all data underlying the findings in their manuscript fully available?

Reviewer #1: Yes

Reviewer #2: Yes

4. Is the manuscript presented in an intelligible fashion and written in standard English?

Reviewer #1: Yes

Reviewer #2: Yes

5. Review Comments to the Author

Reviewer #1: Research is very relevant to scientific knowledge and practical applications. The work is well written, follows a logical sequence, the methodological procedures employed are well detailed, the results are well presented and discussed and the conclusions are consistent with the results obtained.

Here are some recommendations to improve the quality of your work and make it easier for the reader to understand:

1 - The paper should be reviewed to verify and correct some typos or repetition of information.

2 - I suggest that Table 2 be deleted, since the information is already in the body of the text.

3 - I recommend that the tables and figures be inserted as close as possible in the first citation referring to them in the body of the text.

4 - I recommend that table 3 presents the summary of the analysis of variance containing: degrees of freedom, sum of the mean square for the treatments, residues and coefficient of variation.

Reviewer #2: The authors have dealt with a very important topic related to the use of plant tissue analysis as tool for reducing cyanogenic glucoside contents in cassava roots supplied with NPK. In general, this study led to very interesting results but I have few comments:

- I didn't really see how the plant tissue analysis alone as being sufficient to reduce HCN content in root. I see more the plant tissue analysis as a tool to assess HCN and nutrient deficiency diagnostic tools, which do not necessary imply actions towards reducing HCN content or correcting the deficiency. Doing the diagnostic is one thing, taking action towards correcting the deficiency or reducing the HCN content is another one. So, I suggest to revise the title towards something like 'Plant tissue analysis as tool for early diagnostic (or for predicting) HCN contents in roots supplied with NPK fertilizers'.

- Introduction and results (280-395). I have also realized a wrong use of the terms 'balanced NPK fertilizers'. A balanced NPK fertilizers means the nutrients (N, P and K) are in proportions that meet adequately the plant requirements. The paper does not show any information confirming that the quantity of fertilizers used are balanced. With the application of balanced fertilizers, we expect balanced nutrition when growing conditions are met and the plant is expected to respond to increasing supplies of balanced quantities of fertilizers with a maximum nutrient use efficiency up to about 70% of the attainable yield of the given location before the nutrient use efficiency starts to decline. So below this threshold of 70% attainable yield under balanced nutrition conditions, I'd expect HCN content of the plant to remain in reasonable proportions. Under imbalanced nutrition, HCN is likely to increase non proportionally to the plant growth. I suggest then that the authors consider to relate their finding to the level of nutrient use efficiency of the varieties used. They can use either the internal nutrient use efficiency (dry root yield divided by total nutrient uptake) or agronomic efficiency (yield increase due to fertilizer divided by the quantity of fertilizer applied). This will give an indication of how responsive the variety is to fertilizers in relation to HCN content.

- In the discussions, include briefly cost implications and feasibility. Plant tissue analyses are not cheap.

- After doing plant tissue analysis and found HCN content of the roots or leafs, what actions to undertake to reduce it? This is not clearly discuss in the paper.

- How do the results/conclusions of this study relate to roots HCN content at 10-12months after planting and 15months after planting? Given the fact the longer the cassava stays in the field, the higher the root yield, will the HCN content reduce with time?

- Statistical analysis done on yearly basis is justified due to the fact that the soil has been corrected for S, Zn and Cu in the second year. Thus, merge Tables 5 and 6 and show results for each year for all the varieties where applicable.

- Lines 237-255: indicate plant age for all the analyses for both pot and field experiments.

- Lines 317-318: Revise the conversion of 0.54 and 0.51mg/100g. It gives 5.4 and 5.1mg/kg instead of 54 and 51mg/kg.

- Line 416: Is it not Table 5 instead of Table 6?

- Table 7 shows contradictory results, which prevent any generalization of the conclusions of this study.

In view of the above points, I recommend its acceptation after revision of relevant comments.

6. PLOS authors have the option to publish the peer review history of their article (what does this mean?). If published, this will include your full peer review and any attached files.

Reviewer #1: No

Reviewer #2: No

---

## [Author Response · Author response to Decision Letter 0]

26 Nov 2019

The following changes have been made:

The title has been changed as advised, but with a slight twist.

Typos, repetitions were worked on, the work was edited to improve readability.

Complete ANOVA tables have been added.

The term balanced fertilizer has been removed.

Reference has been made to plant age and its effects on root HCN.

Tables have been moved to where they are first mentioned.

Unnecessary tables have been removed.

---

## [Decision Letter · Decision Letter 1]

22 Jan 2020

Plant tissue analysis as a tool for predicting fertiliser needs for low cyanogenic glucoside levels in cassava roots: An assessment of its possible use

PONE-D-19-21677R1

Dear Dr. Imakumbili,

We are pleased to inform you that your manuscript has been judged scientifically suitable for publication and will be formally accepted for publication once it complies with all outstanding technical requirements.

With kind regards,

Paulo H. Pagliari

Academic Editor

PLOS ONE

Additional Editor Comments (optional):

Reviewers' comments:

Reviewer's Responses to Questions

**Comments to the Author**

1. If the authors have adequately addressed your comments raised in a previous round of review and you feel that this manuscript is now acceptable for publication, you may indicate that here to bypass the “Comments to the Author” section, enter your conflict of interest statement in the “Confidential to Editor” section, and submit your "Accept" recommendation.

Reviewer #1: All comments have been addressed

Reviewer #2: All comments have been addressed

2. Is the manuscript technically sound, and do the data support the conclusions?

Reviewer #1: Yes

Reviewer #2: Yes

3. Has the statistical analysis been performed appropriately and rigorously? 

Reviewer #1: Yes

Reviewer #2: Yes

4. Have the authors made all data underlying the findings in their manuscript fully available?

Reviewer #1: Yes

Reviewer #2: Yes

5. Is the manuscript presented in an intelligible fashion and written in standard English?

Reviewer #1: Yes

Reviewer #2: Yes

6. Review Comments to the Author

Reviewer #1: The authors made the main corrections and changes suggested by the reviewers and justified those that could not be accomplish.

Reviewer #2: The authors have made significant improvement to the manuscript. This provides more clarity in the communication of the results of their research.

7. PLOS authors have the option to publish the peer review history of their article (what does this mean?). If published, this will include your full peer review and any attached files.

Reviewer #1: No

Reviewer #2: No

---

## [Editor Report · Acceptance letter]

29 Jan 2020

PONE-D-19-21677R1 

Plant tissue analysis as a tool for predicting fertiliser needs for low cyanogenic glucoside levels in cassava roots: An assessment of its possible use 

Dear Dr. Imakumbili:

I am pleased to inform you that your manuscript has been deemed suitable for publication in PLOS ONE. Congratulations! Your manuscript is now with our production department. 

With kind regards,

on behalf of

Dr. Paulo H. Pagliari 

Academic Editor

PLOS ONE